# Clinical impact of rapid molecular diagnostic tests in patients presenting with viral respiratory symptoms: A systematic literature review

Ali Mojebi[1], Ping Wu[1], Sam Keeping[1], Braden Hale[1], Jordan G. Chase[2]*, Anne Beaubrun[2]

1 Evidence Synthesis, PRECISIONheor, Vancouver, BC, Canada, 2 Global Health Economics & Outcomes Research, Cepheid, Sunnyvale, CA, United States of America

* jordan.chase@cepheid.com

## Abstract

### Background

Molecular tests can detect lower concentrations of viral genetic material over a longer period of respiratory infection than antigen tests. Delays associated with central laboratory testing can result in hospital-acquired transmission, avoidable patient admission, and unnecessary use of antimicrobials, all which may lead to increased cost of patient management. The aim of this study was to summarize comparisons of clinical outcomes associated with rapid molecular diagnostic tests (RMDTs) versus other diagnostic tests for viral respiratory infections.

### Methods

A systematic literature review (SLR) conducted in April 2023 identified studies evaluating clinical outcomes of molecular and antigen diagnostic tests for patients suspected of having respiratory viral infections.

### Results

The SLR included 21 studies, of which seven and 14 compared RMDTs (conducted at points of care or at laboratories) to standard (non-rapid) molecular tests or antigen tests to detect SARS-CoV-2 and influenza, respectively. In studies testing for SARS-CoV-2, RMDTs led to reductions in time to test results versus standard molecular tests (range of the reported medians: 0.2–3.8 hours versus 4.3–35.9 hours), with similar length of emergency department stay (3.2–8 hours versus 3.7–28.8 hours). Similarly, in studies testing for influenza, RMDTs led to reductions in time to test results versus standard molecular tests (1–3.5 hours versus 18.2–29.2 hours), with similar length of emergency department stay (3.7–11 hours versus 3.8–11.9 hours). RMDTs were found to decrease exposure time of uninfected patients, rate of hospitalization, length of stay at the hospitals, and frequency of unnecessary antiviral and antibacterial therapy, while improving patient flow, compared to other tests.

**Data Availability Statement:** The data summarized in this systematic literature review were derived from the individual published studies that were

included in the evidence base. All 21 included studies have been cited with complete references and can be accessed through their respective journals. All data are available within the paper (Tables 1–9) or its Supplementary Information files. Hyperlinks are provided within the paper in the reference list.

**Funding:** AM, PW, SK, and BH are employees of PRECISIONheor, which received funding from Cepheid for this work. Jordan G. Chase and Anne Beaubrun are employees of Cepheid (which provided the funding) and were involved with the study design, data collection, decision to publish, and preparation of the manuscript.

## Conclusions

Compared to other diagnostic tests, RMDTs improve clinical outcomes, test turnaround time, and stewardship by decreasing unnecessary use of antibiotics and antivirals. They also reduce hospital admission and length of stay, which may, in turn, reduce unnecessary exposure of patients to hospital-acquired infections and their associated costs.

## 1. Introduction

The World Health Organization (WHO) estimates 3–5 million new cases of severe influenza each year, resulting in nearly half a million deaths worldwide [1, 2]. As of August 2023, there has also been over 750 million confirmed cases and around seven million deaths related to the COVID-19 pandemic [3]. Early and accurate detection of viral respiratory infections, such as SARS-CoV-2 and influenza, reduces their spread, severity, and duration, leading to reductions in unnecessary healthcare resource utilization, improvements in patient outcomes, and prevention of onward infection [4–6].

Decisions regarding choice of diagnostic tests are made based on the suspected pathogen, time, cost, availability of testing supplies, and patient risk category [7–9]. For example, levels of antibodies rise too slowly against viral pathogens such as the influenza virus (peaking at 6–7 weeks) and SARS-CoV-2 (peaking at two weeks) to be detectable within a sufficiently short time after initiation of the symptoms [10, 11]; therefore, antibody tests will have low sensitivity for, and are not suitable for a timely diagnosis of, acute (i.e., current) infections with the influenza virus and SARS-CoV-2 [12, 13]. Molecular and antigen tests are more reliable alternatives and provide information on current infection and, therefore, are more effective in guiding patient care and treatment decisions [14]. When timely molecular testing is not feasible, antigen testing is recommended for identifying infected individuals [15]. However, the sensitivity and negative predictive value of antigen tests are still heavily dependent on viral load, resulting in suboptimal diagnostic performance for viral infections, such as influenza and SARS-CoV-2, compared to molecular tests (e.g., nucleic acid amplification tests [NAATs]) [7, 16–18]. Conversely, molecular tests can detect lower concentrations of viral material over a longer period of infection than antigen tests and, therefore, have become the "gold standard" for diagnosing respiratory viral infections [13, 19]. As such, guidelines from the Infectious Diseases Society of America (IDSA) and European Society of Clinical Microbiology and Infectious Diseases recommend rapid reverse-transcriptase (RT) polymerase chain reaction (PCR) or laboratory-based NAAT as the testing methods of choice for diagnosing SARS-CoV-2 infections [15, 20].

Rapid molecular diagnostic tests (RMDTs), which can be conducted at the laboratory or at the point of care (PoC), are a type of molecular assay that can yield results in as fast as 15–30 minutes [21]. Over the past two decades, numerous systematic literature reviews (SLRs) and meta-analyses (MAs) have sought to identify and describe evidence on the diagnostic performance of RMDTs for respiratory infections in terms of sensitivity, specificity, and positive and negative predictive values [22–41]. However, fewer SLRs/MAs have summarized and synthesized evidence on the impact of these tests in terms of therapeutic decisions and patient outcomes. Specifically, two pre-COVID-19 SLRs/MAs of studies among patients of all ages drew mixed conclusions regarding the impact of RMDTs and antigen diagnostic tests on the prescription of antibiotics and antivirals and length of stay (LoS) at the emergency department (ED) [39, 42]. However, two post-COVID SLR/MAs of studies among pediatric patients concluded that RMDTs reduced antibiotic prescriptions and increased antiviral prescriptions,

with some evidence suggesting that these tests also reduced LoS at the hospital and duration of therapy [24, 43].

With the rapid expansion of research in this area, particularly since the onset of the COVID-19 pandemic, the current study aimed to provide an updated understanding of the most up to date evidence on the clinical impact of RMDTs compared to standard (i.e., non-rapid) laboratory molecular tests and antigen tests in adults suspected of respiratory viral infections.

## 2. Materials and methods

An SLR was conducted, according to established guidance, with pre-specified study eligibility criteria in terms of populations, interventions, comparators, outcomes, and study designs of interest (Table 1) [44–46]. Studies were eligible for inclusion if they reported clinical outcomes for RMDTs versus standard (i.e., non-rapid) laboratory molecular tests and/or antigen tests for the diagnosis of influenza A virus, influenza B virus, SARS-CoV-2, and/or respiratory syncytial virus (RSV) in adult patients suspected of viral respiratory infections and published in English from 2019 through 2023. Studies exclusively conducted in at-risk populations, such as children, healthcare workers, severely ill patients (including those admitted to intensive care units or oncology wards) and pregnant women, were not included.

Searches were conducted in the Embase, MEDLINE, EconLit, and Cochrane Central Register of Controlled Trials databases, with search strategies (S1–S4 Tables) that included a combination of subject headings and free-text terms for diseases, interventions, and study designs of interest. These were complimented by searches of the past two iterations of relevant conferences (IDWeek, American Thoracic Society International Conference, American Society of Tropical Medicine and Hygiene Annual Meeting, and European Congress of Clinical Microbiology and Infectious Diseases).

During both abstract and full-text screening stages, each record was assessed by a single reviewer. Screening decisions were then verified by a senior reviewer independently. Similarly, for each included study, data were recorded by a single reviewer and quality-checked by the senior reviewer. At each stage, any discrepancies were resolved by discussion between reviewers, with the option to include a third more senior reviewer, if needed. Data were collected for outcomes of interest (Table 1) as well as for study characteristics, baseline patient characteristics, and the analysis method(s) used in the included studies. Quality assessment of the included studies was performed by the senior reviewer using the ROBINS-I tool (for non-randomized studies) and the Cochrane Collaboration risk-of-bias tool for randomized controlled trials [47, 48]. Data were stored and managed in a Microsoft® Excel workbook. The process of study identification and selection were summarized with a Preferred Reporting Items for Systematic Reviews and Meta-Analyses (PRISMA) flow diagram (S1 Checklist) [46]. The study protocol was not previously published or registered.

## 3. Results

### 3.1. Overview of the evidence base

Searches were executed on April 19, 2023, where 10,594 citations were identified from the main databases. After removing 1,907 duplicates and excluding 8,238 abstracts, 449 full-text articles were reviewed of which 22 were included. Two additional citations were identified via searches of other sources, resulting in a total of 24 citations, representing 21 unique studies, being included in the SLR [49–72]. Of the 21 included studies, seven tested patients for SARS-CoV-2 and 14 tested patients for influenza with or without RSV (Fig 1), which are summarized separately in Section 3.2 and Section 3.3, respectively.

**Table 1. Study eligibility criteria of the systematic literature review.**

| Component | Inclusion criteria | Exclusion criteria |
|---|---|---|
| **Population** | Individuals presenting with symptoms of respiratory viral infection | Studies conducted exclusively in specific populations, such as children, pregnant women, healthcare workers, and severely ill patients |
| **Interventions** | Rapid[a] molecular[b] diagnostic tests, conducted at laboratory or at point of care[c], for one or more of the following viruses, administered in an outpatient or inpatient clinical setting:<br>• Influenza A virus<br>• Influenza B virus<br>• SARS-CoV-2<br>• RSV | -- |
| **Comparators** | • Standard, non-rapid, laboratory molecular tests<br>• Antigen tests[d] | -- |
| **Outcomes** | • Length of stay at the ED<br>• Length of time under medical observation<br>• Admission to hospital<br>• Length of stay at the hospital<br>• Ancillary testing (e.g., radiography, ultrasound)<br>• Antimicrobial prescription in patients with negative test results | -- |
| **Study Design** | • Randomized controlled trials<br>• Non-randomized controlled trials<br>• Comparative (multicohort) prospective and retrospective observational studies | • Non-comparative studies (e.g., single-arm trials, single-cohort observational studies)<br>• Cost-effectiveness analyses<br>• Economic modeling studies<br>• Animal or *in vitro* studies<br>• Case series/case reports<br>• Cross-sectional studies<br>• Editorials, commentaries, letters, reviews<br>• Systematic reviews<br>• Meta-analyses |
| **Time** | Studies published from 2019 to 2023 | |
| **Language** | English | |

[a]Rapid molecular tests were defined as molecular tests with results available in <3 h

[b]Molecular tests were defined as tests that detect genetic material from a virus using a nucleic acid amplification technique such as reverse transcription (RT) PCR, isothermal amplification (e.g., RT-recombinase polymerase amplification, RT loop-mediated isothermal amplification, transcription-mediated amplification, nicking enzyme-assisted reaction, clustered regularly interspace short palindromic repeats, or next-generation sequencing) [13, 14].

[c]Point-of-care tests were defined as tests administered at or near the site of patient care (e.g., bedside, clinician's office, emergency department)

[d]Antigen tests were defined as tests that detect proteins from a virus using a technique such as enzyme-linked immunosorbent assay, chemiluminescence immunoassay, lateral flow immunochromatographic assay, or lateral flow assay [13, 14].

ED, emergency department; ICU, intensive care unit; RSV, respiratory syncytial virus.

Almost all included studies (20/21) were non-randomized in design, where there was a risk that the compared cohorts were different in terms of the distribution of important baseline patient characteristics that were prognostic of the evaluated outcomes and could act as confounders. As such, the only concern regarding risk of bias was that around half of the studies did not employ an appropriate analysis method to quantitatively control for the above-mentioned differences and did not report sufficient data on baseline patient characteristics (precluding a qualitative comparison of cohorts). Beyond this, there was no major concern regarding the other evaluated domains, e.g., missing data (S1 and S2 Figs).

## 3.2. Studies testing for SARS-CoV-2

**3.2.1. Overview of included studies.** Seven studies were included in this category. Brendish et al, 2020 was a prospective, non-randomized controlled study comparing time to test

**Searches executed on April 19, 2023**

**Fig 1. Study selection flow diagram of the systematic literature review.**

results, LoS at the ED and at the hospital, and rate of ancillary tests in adults presenting with COVID-19 symptoms to the ED or other acute medical wards, receiving PoC RMDT versus standard molecular test, using Chi-Square test, independent-samples t tests or Mann-Whitney U tests. Cancella de Abreu et al, 2023 was a retrospective study in the ED of multiple hospitals, comparing time to test results, hospitalization rates and LoS at the ED between patients receiving standard molecular tests (March-May 2020) and those receiving RMDT (October-

December 2020). Collier et al, 2020 comprised of a clinical validation study and a clinical implementation study. In the latter part, patients undergoing SARS-CoV-2 testing in a 10-day period before (standard molecular test) and after (RMDT) introduction of RMDT were compared in terms of LoS at the hospital and time to test results using Wilcoxon rank sum tests. Gerlier et al, 2021 was a prospective non-randomized before-after trial comparing LoS at the ED and time to test results in patient receiving standard molecular test versus those receiving RMDT recruited from two consecutive seven-week periods, using Mann-Whitney or Kruskal-Wallis tests. Hinson et al, 2021 was a retrospective cohort study of patients presenting to the ED, where uninfected patient exposure time was compared between patients receiving RMDTs (those expected to be hospitalized or could not be discharged to self-isolate at home) and patients receiving standard molecular tests, using boxplot analysis and log-rank test for time-interval data. Livingstone et al, 2022 was a pre- and post-implementation study that compared LoS at the ED and time to test results in patients receiving standard molecular test (March 1-August 13, 2020) versus those receiving PoC RMDT (August 14, 2020-April 1, 2021), using the Mann-Whitney U test. Lastly, in Mortazavi et al, 2022, patients presenting to the ED of a single center were studied across three distinct time periods separated by the introduction of RADTs and RMDTs (Period 1: standard molecular test; Period 2: RADT followed by standard molecular test when RADT was negative; and Period 3: RADT followed by RMDT when RADT was negative). However, patients were not required to have tested for SARS-CoV-2 at study entry. As a result, many patients in each period did not test for SARS-CoV-2 or had tested positive before admission to the ED. Patients also did not always follow the above-mentioned protocols; for example, around 10% of patients in Period 3 underwent standard molecular tests. LoS at the ED and at the hospital, as well as rate of hospitalization were compared across the three periods, using one-way-ANOVA with Tukey's multiple comparison tests and Fisher's exact test.

Overall, two studies were non-randomized controlled trials, two were prospective implementation studies, and three were retrospective in design. Four studies were conducted in a single center, whereas three were carried out in multiple centers. All studies were conducted in Europe except Hinson et al, 2021, which was conducted in North America. All studies were conducted in patients presenting with respiratory symptoms and suspected of having a SARS-CoV-2 infection, except for Cancella de Abreu et al, 2023, where only those with a confirmed positive SARS-CoV-2 test were eligible (Table 2). Sample sizes ranged from 1,054 to 3,333, except for Livingstone et al, 2022 and Hinson et al, 2021, with 6,628 and 9,018 patients, respectively.

Six studies compared RMDTs to standard (i.e., non-rapid) molecular tests and one (Mortazavi et al, 2022) compared RMDTs to rapid antigen detection tests (RADTs). In all studies, RMDTs were conducted at the PoC, except Hinson et al, 2021, where they were conducted at a laboratory. Standard molecular tests were PCR in all studies; RMDTs were PCR in all studies except Collier et al, 2020 and Gerlier et al, 2021, where isothermal NAATs were used (Table 3).

Few baseline patient characteristics were reported in the included studies. In Brendish et al, 2020, around 20%, 10%, 40%, and 15% of patients had diabetes mellitus, renal diseases, hypertension, and chronic obstructive pulmonary disease, respectively. In Gerlier et al, 2021, around 20% of patients had cardiovascular diseases. Livingstone et al, 2022 reported that around 10%, 10%, and 20% of patients had diabetes mellitus, ischemic heart disease, and chronic obstructive pulmonary disease, respectively, with almost a third of the population having hypertension. Median age and mean age ranged from 68 to 75.2 years (three studies) and from 61 to 66 years (three studies), respectively. In Hinson et al, 2021, 34.1% of the population were over 65 years old. Sex was evenly distributed, with males comprising 45.5% to 64% of the populations (Table 4).

**3.2.2. Time to test results.**   Median time to test results were reported in six studies, ranging from 0.2 to 3.8 hours for RMDTs and from 4.3 to 35.9 hours for standard molecular tests.

**Table 2. Characteristics and patient eligibility criteria of studies testing for SARS-CoV-2.**

| Publication | Study design | Region/centers | Setting | Eligibility criteria | Diagnosis status |
|---|---|---|---|---|---|
| **Brendish et al, 2020** [49] | Non-randomized controlled trial | UK, single center | Acute medical unit, ED, or other acute areas | • Age ≥18 years<br>• Acute respiratory illness, or<br>• Without acute respiratory illness but suspected to have COVID-19<br>• Could be recruited within 24 hours of presentation | Suspected |
| **Cancella de Abreu et al, 2023** [50] | Retrospective study | France, multi-center | ED | • Age >16 years<br>• With COVID-19 symptoms and positive SARS-CoV-2 test with a time recorded in the medical chart | With confirmed diagnosis |
| **Collier et al, 2020** [51] | Prospective implementation study | UK, multi-center | ED or acute medical assessment unit | • Age >16 years<br>• Possible case of SARS-CoV-2 infection | Suspected |
| **Gerlier et al, 2021** [52] | Non-randomized trial | France, single center | ED | • Clinical suspicion of moderate or severe COVID-19, or<br>• Requiring urgent surgery or hospitalization | Suspected or screening (for urgent surgery) |
| **Hinson et al, 2021** [53] | Retrospective study | US, multi-center | ED | • Age ≥18 years<br>• Had a laboratory diagnostic evaluation for SARS CoV-2 infection initiated during their ED stay, and<br>• Remained in the hospital (ED or inpatient) until their test results were included | Suspected or screening (for urgent surgery) |
| **Livingstone et al, 2022** [54] | Prospective implementation study | UK, single center | ED (the acute medical unit) | • Tested for SARS-CoV-2 with laboratory or PoC molecular test | Suspected or screening (for urgent surgery) |
| **Mortazavi et al, 2022** [55] | Retrospective study | Sweden, single center | ED | • With COVID-19 symptoms | Suspected or screening |

ED, emergency department; PoC, point-of-care; UK, United Kingdom; US, United States.

All six studies found significant reductions in time to test results with RMDT compared to standard molecular tests.

**3.2.3. Antimicrobial prescription in patients with a negative test.** Antimicrobial prescriptions in patients with negative test was not reported in any of the included studies.

**3.2.4. SARS-CoV-2 test positivity.** SARS-CoV-2 test positivity was often similar in the RMDT and standard molecular test groups, except in Brendish et al, 2020, where it was significantly higher in those undergoing RMDT compared to a standard molecular test (39.5% versus 27.9%, p = 0.0001).

**3.2.5. Length of stay in an emergency department.** Median LoS at the ED in all-comers, reported in four studies, ranged from 3.2 to 8 hours in patients undergoing RMDT and from 3.7 to 28.8 hours in patients undergoing standard molecular tests. In Cancella de Abreu et al, 2023 (all positive patients), median LoS at the ED was 7.6 hours versus 20.6 hours (p<0.001) among hospitalized patients undergoing RMDT and standard molecular tests, respectively. In Mortazavi et al, 2022, mean LoS at the ED decreased by 15 minutes (95% confidence interval [CI]: 7.6–37.6) from Period 2 to Period 3 (i.e., after the introduction of RMDT) in patients who tested positive at the ED.

**3.2.6. Admission to hospital.** Patients undergoing RMDT were less likely to be hospitalized compared to those receiving standard molecular tests in Cancella de Abreu et al, 2023 (81.7% versus 86.3%, p<0.001). In Mortazavi et al, 2022, however, no statistically significant difference in hospital admission rate was observed between RMDT (50.5% [Period 3]) and standard molecular tests (51.6% [Period 2]). Of note, Mortazavi et al, 2022 reported that

**Table 3. Diagnostic tests evaluated in studies testing for SARS-CoV-2.**

| Publication | Test summary | Details |
|---|---|---|
| **Brendish et al, 2020** | RMDT (at the PoC) | QIAstat-Dx Respiratory SARS-CoV-2 Panel (singleplex PCR) |
| | Standard molecular test | PHE Lab PCR (singleplex PCR) |
| **Cancella de Abreu et al, 2023** | RMDT (at the PoC) | QIAstat-Dx Respiratory SARS-CoV-2 Panel (singleplex PCR) |
| | Standard molecular test | SARS-CoV-2 Cobas assay Lab PCR (singleplex PCR) |
| **Collier et al, 2020** | RMDT (at the PoC) | SAMBA II SARS-CoV-2 (isothermal NAAT) |
| | Standard molecular test | Lab PCR (singleplex PCR) |
| **Gerlier et al, 2021** | RMDT (at the PoC) | ID NOW COVID-19 (isothermal NAAT) |
| | Standard molecular test | SimplexaCOVID-19 Direct assay Lab PCR (singleplex PCR) |
| **Hinson et al, 2021** | RMDT (at the PoC) | Xpert Xpress SARS-CoV-2 test (singleplex PCR) |
| | Standard molecular test | RealStar SARS-CoV-2 RT-PCR Kit 1.0 (singleplex PCR) |
| **Livingstone et al, 2022** | RMDT (at the PoC) | FilmArray respiratory panel RP2.1 (singleplex PCR) |
| | Standard molecular test | Lab PCR (singleplex PCR) |
| **Mortazavi et al, 2022** | Standard molecular test (Period 1) | SARS-CoV-2 RT-PCR Lab PCR (singleplex PCR) |
| | RADT followed by standard molecular test (if RADT is negative) (Period 2) | Clinitest RT (SARS-COV-2 antigen, RAT) followed by non-rapid PCR in pts with RAT negative (singleplex antigen) |
| | RADT followed by RMDT (if RADT is negative) (Period 3) | Clinitest RT (SARS-COV-2 antigen, RAT) followed by PoC VitaPCR in pts with RAT negative (singleplex antigen) |

NAAT, nucleic acid amplification test; PCR, polymerase chain reaction; PoC, point-of-care; POCT, point-of-care testing; RAT, rapid antigen test; RADT, rapid antigen diagnostic test; RMDT, rapid molecular diagnostic test.

admission of participants with negative test at the ED was significantly reduced after the introduction of RMDT (25.7% [Period 3] versus 31.2% [Period 2]).

**3.2.7. Length of stay in hospital.** In Brendish et al, 2020, median LoS at the hospital was longer in patients undergoing RMDTs compared to those undergoing standard molecular tests (5.1 versus 4.2 days; p = 0.017), which the study investigators associated with the higher prevalence of SARS-CoV-2 infection in the former cohort. Mortazavi et al, 2022 did not find a significant difference in the mean LoS at hospital between RMDT (6.7 days [Period 3]) and standard molecular tests (6.0 days [Period 2]); however, the mean LoS in patients with a negative test at the ED was shorter with RMDT (Period 3) compared to standard molecular tests (Period 2) (5.1 versus 5.8 days; p = 0.046). Lastly, Collier et al, 2020 reported a shorter median LoS at hospital with RMDT compared to standard molecular tests (2.9 versus 4.4 days; p<0.0001).

**3.2.8. Ancillary testing.** Ancillary testing was only reported in Brendish et al, 2020, where patients undergoing RMDT were 7% (95% CI: 4%-9%) less likely to receive a chest X-ray compared to those undergoing standard molecular tests.

**3.2.9. Length of stay under medical observation.** In Collier et al, 2020, the median time from admission to definitive bed placement was shorter in those undergoing RMDT compared to those undergoing a standard molecular test (17.1 versus 23.4 hours; p = 0.02). In Hinson et al, 2021, the median exposure time of uninfected patients (defined as the time from test order to first treatment space re-assignment after a negative result) was shorter in those undergoing RMDT compared to those undergoing a standard molecular test (6.6 versus 19.2 hours; p<0.001) (Table 5).

**Table 4. Baseline patient characteristics of studies testing for SARS-CoV-2.**

| Publication | Test | N | Age, median (IQR) | Male, n (%) | Comorbidities, n (%) |
|---|---|---|---|---|---|
| **Brendish et al, 2020** | RMDT (at the PoC) | 499 | 68 (51–81) | 262 (53.0) | Reported for 475 patients<br>• Diabetes mellitus: 108 (23.0)<br>• Renal disease: 38 (8.0)<br>• Hypertension: 175 (37.0)<br>• COPD: 93 (19.0)<br>• Cancer: 40 (8.0) |
| | Standard molecular test | 555 | 70 (51–81) | 303 (55.0) | Reported for 554 patients<br>• Diabetes mellitus: 135 (24.0)<br>• Renal disease: 85 (15.0)<br>• Hypertension: 247 (45.0)<br>• COPD: 85 (15.0)<br>• Cancer: 36 (6.0) |
| **Cancella de Abreu et al, 2023** | RMDT (at the PoC) | 329 | 66 (18.0)[a] | 189 (57.4) | -- |
| | Standard molecular test | 1009 | 65 (17.3)[a] | 646 (64.0) | |
| **Collier et al, 2020** | RMDT (at the PoC) | 799 | 61 (36–78)[b] | 364 (45.6) | -- |
| | Standard molecular test | 388 | 63 (42–79.5)[b] | 197 (50.8) | |
| **Gerlier et al, 2021** | RMDT (at the PoC) | 1856 | 74 (59–84) | 854 (46.0) | • Cardiovascular: 416 (22.4) |
| | Standard molecular test | 1477 | 70 (46–85) | 672 (45.5) | • Cardiovascular: 258 (17.5) |
| **Hinson et al, 2021** | RMDT (at the PoC)/ standard molecular test | 9018 | -- | 4565 (50.6) | -- |
| **Livingstone et al, 2022** | RMDT (at the PoC) | 4640 | 73.9 (54–84.8) | 2527 (54.5) | Charlson Comorbidity Index: 3.1 (1.2–5.0)<br>• Diabetes mellitus: 340 (7.3)<br>• Chronic kidney disease: 27 (0.6)<br>• Ischaemic heart disease: 417 (9.0)<br>• Hypertension: 1354 (29.2)<br>• COPD: 447 (10.3) |
| | Standard molecular test | 1988 | 75.2 (57.5–85.3) | 1023 (51.5) | Charlson Comorbidity Index: 3.7 (1.7–5.7)<br>• Diabetes mellitus: 198 (10.0)<br>• Chronic kidney disease: 15 (0.8)<br>• Ischaemic heart disease: 272 (13.7)<br>• Hypertension: 861 (43.3)<br>• COPD: 332 (16.7) |
| **Mortazavi et al, 2022** | Standard molecular test (Period 1) | 781 | 61 (22.0)[a] | 364 (46.6) | -- |
| | RADT followed by standard molecular test (if RADT is negative) (Period 2) | 988 | 66 (18.0)[a] | 499 (50.5) | |
| | RADT followed by RMDT (if RADT is negative) (Period 3) | 1171 | 61 (20.0)[a] | 580 (49.5) | |

[a]Mean and standard deviation were reported

[b]Mean and IQR were reported

COPD, chronic obstructive pulmonary disease; IQR, interquartile range; PoC, point-of-care; RADT, rapid antigen diagnostic test; RMDT, rapid molecular diagnostic test.

## 3.3. Studies testing for influenza virus

**3.3.1. Overview of included studies.** Fourteen studies were included in this category. Au Yeung et al, 2021 was a retrospective study of patients with confirmed diagnosis at the ED and inpatient hospital settings comparing RMDT to standard molecular test in terms of test turn-around time, admission rates, and LoS at the ED and at the hospital, using Mann-Whitney and Chi-squared tests. Benirschke et al, 2019 was a retrospective study comparing antimicrobial prescribing patterns (when a result was available at the time of visit) with PoC RMDT in one urgent care location to urgent care centers that used RADT (negative specimens reflexed to PCR), using Chi-squared tests. Berry et al, 2020 was a prospective interrupted 'on-off' study in

**Table 5. Outcomes reported in studies testing for SARS-CoV-2.**

| Publication | Patient group | Test | N | Positive tests, n (%), p-value | LoS at the ED, hr, median (IQR) | Admission to the hospital, n (%) | LoS at the hospital, hr, median (IQR) | Ancillary testing, n (%) | Time to test results, hr median (IQR) |
|---|---|---|---|---|---|---|---|---|---|
| **Brendish et al, 2020** | Overall | RMDT (at the PoC) | 499 | 197 (39.5) | 8.0 (6.0–15.0)[a] | -- | 122.4 (48.0–220.8)[b] | Chest X-ray: 488 (98.0)[c] | 1.7 |
| | | Standard molecular test | 555 | 155 (27.9), p = 0.0001 | 28.8 (23.5–38.9)[a], p<0.0001 | -- | 100.8 (28.8–230.4)[b], p = 0.017 | Chest X-ray: 507 (91.0) | 21.3, p<0.0001 |
| **Cancella de Abreu et al, 2023** | Overall | RMDT (at the PoC) | 329 | 329 (100) | 7.1 (4.8–13.1) | 269 (81.7) | -- | -- | 3.1 (1.5–5.8) |
| | | Standard molecular test | 1009 | 1009 (100) | 18.1 (8.7–26.2), p<0.001 | 871 (86.3), p<0.001 | | -- | -- | 10.3 (5.1–16.7), p<0.0001 |
| | Hospitalized patients | RMDT (at the PoC) | 150 | -- | 7.6 (5.1–17.1) | -- | -- | -- | -- |
| | | Standard molecular test | 763 | -- | 20.6 (11.3–26.9), p<0.001 | -- | -- | -- | -- |
| **Collier et al, 2020** | Overall | RMDT (at the PoC) | 799 (913 tests) | 39 (4.3) | -- | -- | 69.6 (21.6–175.2)[b] | -- | 3.8 (2.7–6.0) |
| | | Standard molecular test | 388 (561 tests) | 49 (8.7) | -- | -- | 105.6 (26.4–259.2)[b], p<0.0001 | -- | 35.9 (23.8–48.6), p<0.0001 |
| **Gerlier et al, 2021** | Overall | RMDT (at the PoC) | 1856 | 195 (10.5) | 7.2 (4.4–9.5) | -- | -- | -- | 0.17 (0.17–0.18)[d] |
| | | Standard molecular test | 1477 | 136 (9.2) | 6.7 (4.6–9.2), p = 0.43 | -- | -- | -- | 4.3 (3.2–6.3)[d] |
| **Hinson et al, 2021** | Overall | RMDT (at the PoC) | 3,502 | -- | -- | -- | -- | -- | 1.9 (1.4–2.8) |
| | | Standard molecular test | 5,516 | -- | -- | -- | -- | -- | 7.8 (3.7–11.7), p<0.001 |
| **Livingstone et al, 2022** | Overall | RMDT (at the PoC) | 4640 | -- | 3.2 (2.0–5.6)[e] | -- | -- | -- | 1.0 (0.8–1.3) |
| | | Standard molecular test | 1988 | -- | 12.0 (4.8–20.6)[e], p<0.0001 | -- | -- | -- | 6.5 (2.1–17.9) |
| **Mortazavi et al, 2022** | Overall | Standard molecular test (Period 1) | 781 | 70 (9.0) | 6.4 (4.4)[d,f] | 386 (49.4) | 120 (285.6)[b,f] | -- | -- |
| | | RADT followed by standard molecular test (if RADT is negative) (Period 2) | 988 | 135 (13.7) | 6.3 (4.5)[d,f] | 510 (51.6) | 144 (211.2)[b,f] | -- | -- |
| | | RADT followed by RMDT (if RADT is negative) (Period 3) | 1171 | 203 (17.3) | 6.1 (4.4)[d,f], p = 0.22 | 591 (50.5) | 160.8 (352.8)[b,f], p = 0.1 | -- | -- |
| | SARS-CoV-2 positive | Standard molecular test (Period 1) | 70 | 70 (100) | 6.6 (4.0)[d,f] | 50 (8.4) | 204 (177.6)[b,f] | -- | -- |
| | | RADT followed by standard molecular test (if RADT is negative) (Period 2) | 135 | 135 (100) | 6.1 (3.5)[d,f,g] | 85 (8.6) | 223.2 (218.4)[b,f] | -- | -- |
| | | RADT followed by RMDT (if RADT is negative) (Period 3) | 203 | 203 (100) | 5.8 (3.8)[d,f,g], p = 0.0002 | 122 (10.4) | 192 (264)[b,f], p = 0.08 | -- | -- |
| | SARS-CoV-2 negative | Standard molecular test (Period 1) | 374 | 0 (0.0) | 7.2 (4.6)[d,f] | 255 (32.7) | 158.4 (196.8)[b,f] | -- | -- |
| | | RADT followed by standard molecular test (if RADT is negative) (Period 2) | 515 | 0 (0.0) | 7.4 (5.1)[d,f] | 308 (31.2) | 139.2 (165.5)[b,f] | -- | -- |
| | | RADT followed by RMDT (if RADT is negative) (Period 3) | 569 | 0 (0.0) | 7.1 (4.6)[d,f], p = 0.31 | 301 (25.7) | 122.4 (192)[b,f], p = 0.01 | -- | -- |

[a] Assessed in patients admitted for >24 h; definitive clinical area refers to a designated COVID-19-positive or COVID-19-negative ward

[b] Calculated from days to hours, assuming one day = 24 hours

[c] Difference: 7% (95% CI: 4% to 9%)

[d] Calculated from minutes to hours, assuming one hour = 60 minutes

[e] Length of time spent in the Acute Medical Unit (AMU) assessment area

[f] Reported as mean (standard deviation)

[g] Mean LoS at the ED decreased by 15 minutes (95% confidence interval [CI]: 7.6–37.6) from Period 2 to Period 3 (i.e., after the introduction of RMDT) in patients who tested positive at the ED.

ED, emergency department; hr, hour; IQR, interquartile range; LoS, length of stay; PoC, point-of-care; RADT, rapid antigen diagnostic test; RMDT, rapid molecular diagnostic test.

adults admitted to a respiratory assessment unit, comparing PoC RMDT with standard molecular test in terms of time to patient isolation, LoS and turnaround time from admission to test results. Means of the groups were compared in an independent t-test, and LoS analysis was performed via a linear regression model adjusting for Charlson co-morbidity index score. In Berwa et al, 2022, adults with influenza-like illness in one ED were retrospectively included over three epidemic seasons (2016–2017 to 2018–2019). Rate of prescription of antimicrobials for respiratory infections at the ED was compared between PoC RMDT (implemented in 2018–2019) and standard molecular test (previous seasons), along with prescriptions of chest X-rays, hospitalizations and LoS at the ED, using the Kruskale-Wallis test and Chi-squared test. In Bibby et al, 2022, patients with a respiratory viral test order were randomized, on alternating days, to RMDT at the laboratory followed by standard testing or standard molecular test. Clinicians and patients were blinded to the randomization plan. Prescription rates of antimicrobials and chest X-ray, as well as LoS at the ED, were compared between the two groups among patients tested while in the ED who were admitted to hospital, using Fisher's exact test, unpaired t-test or Mann–Whitney U test. Of note, the majority of chest X-ray orders and oseltamivir prescriptions occurred after the RMDT results were reported. Brooke-Pearce et al, 2019 was a retrospective cohort study to determine the impact of PoC RMDT on the accurate and timely diagnosis of influenza and operational workflow of the medical center during the winter of 2017–2018. No formal hypothesis testing was reported in that publication. Lankelma et al, 2019 was a retrospective study to describe the use of PoC RMDT for patients presenting with symptoms of viral respiratory infection at the ED (2017–2018) in comparison with the previous epidemic (2016–2017), where standard molecular test had been used. Use of antibiotics and use of oseltamivir following test results were analyzed using the Mann-Whitney U-test. Martinot 2019 was a retrospective study in adults who had a confirmed diagnosis with either RMDT or standard molecular test at the ED during the 2017–2018 epidemic. Various outcomes were compared in patients who tested positive only, including LoS at the ED, hospitalization rates and duration of hospitalization. using chi-square and exact Mann-Whitney tests. Melhuish 2020 was another retrospective study of consecutive adult patients presenting to the ED and receiving a PoC RMDT in comparison with those presenting to the ED during another period who were tested by standard molecular test. The main objective of the study was cost comparison between the two cohorts; however, clinical outcomes such as rate and duration of hospitalization were also evaluated. Peaper 2019 was a pre-post study comparing LoS at the ED, rates of empiric oseltamivir prescriptions in patients without influenza, and rates of influenza infection between the 2016–2017 season (RADT, direct fluorescent antigen testing, or standard molecular test) and the 2017–2018 season (RMDT), using chi-square and Mann-Whitney U tests. Wabe et al, 2019a was a controlled quasi-experimental study, where LoS at hospital was compared between patients receiving standard molecular tests during the pre-implementation period (July to December 2016) and those receiving RMDT during the post-implementation period (July to December 2017) using a median regression adjusting for age, study hospital, the source of referral, intensive care admission status, mode of separation, Charlson comorbidity index and type of principal diagnosis. Wabe et al, 2019b was a before-and-after study in consecutive patients tested by standard molecular test during July-December 2016, and in those tested by RMDT during July-December 2017. Hospital admissions, LoS at the ED and test turnaround time were compared between the two periods using logistic regression and quantile regression for binary and continuous outcomes, respectively, adjusting for age and sex, as well as time, day, and mode of arrival at the ED. Wesolowski et al, 2023 was a retrospective cohort study of patients discharged from the ED with a confirmed diagnosis of influenza using standard molecular test (January 2017 - July 2019) or RMDT (July 2019 - February 2020). LoS at the ED as well as rates of hospitalization and ancillary tests were compared

between the two groups, using Chi-Square test, Student's t-test or Mann-Whitney U test. Lastly, Yin et al, 2022 was a prospective study, where physicians' intentions with regard to admission and use of antimicrobials were compared before versus after performing RMDTs and before versus after performing RADT, using the McNemar-Mosteller exact test.

Overall, one study was a randomized controlled trial, two were prospective implementation studies, and 11 were retrospective in design. Ten studies were conducted in a single center, whereas four were carried out in multiple centers. Four studies were conducted in North America, seven in Europe, and three in Australia. Eleven studies were conducted in patients presenting with respiratory symptoms and suspected of having an influenza infection. Among these studies, Wabe et al, 2019a qualified patients hospitalized for a respiratory illness with at least one laboratory test result, and Yin et al, 2022 included patients requiring admission or suffering from an underlying condition at risk of respiratory complications. In the remaining three studies (Au Yeung et al, 2021, Wesolowski et al, 2022 and Martinot et al, 2019), only those with a confirmed positive influenza test were eligible (Table 6). Sample sizes generally ranged from 178 to 2,162, except for Wabe et al, 2019b and Peaper et al, 2021, with 3,741 and 5,118 patients, respectively.

Eleven studies compared RMDT to standard molecular tests, while the remaining three compared RMDT to RADT followed by a standard molecular test if RADT was negative (Benirschke et al, 2019), to RADT, a standard molecular test, or both (Peaper et al, 2021), and to RADT followed by viral culture (Yin et al, 2022). All studies tested for both influenza virus A and influenza virus B, except Wesolowski et al, 2023, which did not mention the type of influenza virus tested. Lastly, the diagnostic assays used in nine studies also tested for RSV (i.e., in addition to influenza). RMDTs were conducted at the PoC in six studies, at a laboratory in five studies, and at either a laboratory or the PoC in one study (Yin et al, 2022); Martinot et al, 2019 and Wesolowski et al, 2023 did not explicitly mention the exact site where RMDTs were conducted. Standard molecular tests were PCR in all studies; RMDTs were PCR in all studies except in Martinot et al, 2019, where isothermal NAATs were used (Table 7).

Few baseline patient characteristics were reported in the studies included in this category. In Melhuish et al, 2020, around 90% of the population had comorbidities, including chronic lung diseases in almost half of the population. Just under 10% of the adult patients in Wesolowski et al, 2023 had asthma, with a small number of patients reported to have other comorbidities (e.g., diabetes mellitus, coronary artery disease). Median age and mean age ranged from 53 to 81 years (six studies) and from 29.2 to 66 years (three studies), respectively. In Bibby et al, 2022 and Yin et al, 2022, 47.4% and 33.8% of patients were over 65 years old, respectively. Sex was evenly distributed, with males comprising 37.5% to 59.8% of the populations (Table 8).

**3.3.2. Time to test results.** Time to test results was reported in 10 studies. Among the studies comparing RMDTs versus standard molecular tests, median time to test results ranged between 1 and 3.5 hours with RMDTs and between 18.2 and 29.2 hours with standard molecular tests. Mean time to test results were reported in Berry et al, 2020 and Wesolowski et al, 2023, where RMDTs resulted in significantly shorter mean turnaround times (2.9 and 3.5 hours, respectively) compared to standard molecular tests (31.2 and 27 hours, respectively). Benirschke et al, 2019 reported the median time to test results as 29 minutes, 16 minutes, and 21 hours for RMDT, RADT, and standard molecular tests (in RADT-negative patients), respectively. Lastly, in Peaper et al, 2021, median time to test results in patients undergoing RMDT was significantly shorter than that of the control group receiving RADT and/or a standard molecular test (2.4 versus 9.9 hours)

**3.3.3. Antimicrobial prescription in patients with a negative test.** Two studies found a significant reduction in antibiotic prescription rates with RMDT compared to standard

**Table 6. Characteristics and patient eligibility criteria of studies testing for influenza virus.**

| Publication | Study design | Region/centers | Setting | Eligibility criteria | Diagnosis status |
|---|---|---|---|---|---|
| **Au Yeung et al, 2021 [56]** | Retrospective study | Australia, single center | ED or in-patient | • Age ≥18 years<br>• Only those with positive RPCR or standard MPCR | With confirmed diagnosis |
| **Benirschke et al, 2019 [57]** | Retrospective study | US, single center | Urgent care setting | • Patients that visited urgent care centers<br>• Tested for influenza during this period | Suspected |
| **Berry et al, 2020 [58]** | Prospective implementation study | UK, single center | Adult respiratory assessment unit | • Adults presenting to hospital with symptoms of an ARTI | Suspected |
| **Berwa et al, 2022 [59]** | Retrospective pre/post implementation study | France, single center | ED | • Age >16 years<br>• Patients were eligible for participation if they presented ILI<br>• The criteria for ILI included fever or feverishness and at least one of the following symptoms: sore throat, cough, myalgia or headache<br>• These patients were all tested for influenza A and B | Suspected |
| **Bibby et al, 2022 [60]** | Randomized controlled trial | Canada, single center | ED or in-patient | • Patients with a respiratory viral testing order were randomized | Suspected |
| **Brooke-Pearce and Demertzi, 2019 [61]** | Retrospective study | UK, single center | Acute 500-bed hospital | • These consisted of flu like symptoms or temperature of ≥ 38˚ before or on presentation to ED<br>• Acute onset of at least one of the following respiratory symptoms: cough with or without sputum; hoarseness; nasal discharge or congestion; shortness of breath; sore throat; wheezing; or sneezing | Suspected |
| **Lankelma et al, 2019 [62]** | Retrospective study | Netherlands, single center | ED | • Patients with acute respiratory tract infection presenting at the ED | Suspected |
| **Martinot et al, 2019 [63]** | Retrospective study | France, single center | ED | • Age ≥18 years<br>• Diagnosed with a molecular flu test (Alere-i Influenza A/B or classic RT-PCR) in the ED | With confirmed diagnosis |
| **Melhuish et al, 2020 [64]** | Retrospective study | UK, single center | ED | • Age >16 years<br>• Patient with symptoms of influenza (fever, plus 2 or more of cough, sore throat, headache, rhinorrhea, myalgia, vomiting and diarrhea) or with a clinical diagnosis of pneumonia, lower respiratory tract infection or infective exacerbation of COPD | Suspected |
| **Peaper et al, 2021 [65]** | Retrospective pre/post implementation study | US, multi-center | ED | — | Suspected |
| **Wabe et al, 2019a [66]** | Retrospective pre/post implementation study | Australia, multi-center | In-patient | • Age ≥18 years<br>• Admitted with a respiratory illness during the study period<br>• Having at least one laboratory test result | Suspected |
| **Wabe et al, 2019b [67]** | Retrospective pre/post implementation study | Australia, multi-center | ED | • All patients tested for influenza virus or RSV | Suspected |
| **Wesolowski et al, 2023 [68]** | Retrospective study | US, single center | ED | • Patients with positive influenza results | With confirmed diagnosis |
| **Yin et al, 2022 [69]** | Prospective implementation study | Belgium, multi-center | ED | • Either a pre-test indication of hospitalization or<br>• An underlying situation at risk of respiratory complication following influenza infection as described by the European Centre for Disease Prevention and Control and the US Centers for Disease Control | Suspected |

ARTI, acute respiratory tract infection; COPD, chronic obstructive pulmonary disease; ED, emergency department; ILI, influenza-like illness; MPCR, standard multiplex polymerase chain reaction; POCT, point-of-care testing; RPCR, rapid nucleic acid testing; RSV, respiratory syncytial virus; RT-PCR, real-time polymerase chain reaction; UK, United Kingdom; US, United States.

**Table 7. Diagnostic tests evaluated in studies testing for influenza virus.**

| Publication | Test summary | Details[a] |
|---|---|---|
| **Au Yeung et al, 2021** | RMDT (at the PoC) | Cepheid Xpert Xpress Flu/RSV Assay (multiplex PCR) |
| | Standard molecular test | Allplex Respiratory Panel kit (highplex PCR) |
| **Benirschke et al, 2019** | RMDT (at the PoC) | Cobas Liat Influenza A/B assay (multiplex PCR) |
| | RADT followed by standard molecular test (if RADT is negative) | Quidel QuickVue Influenza A+B antigen test followed by Simplexa Flu A/B RSV assay for negative pts (antigen) |
| **Berry et al, 2020** | RMDT (at the PoC) | GeneXpert Xpress Flu A and B/RSV analyser (multiplex PCR) |
| | Standard molecular test | Lab PCR (highplex PCR) |
| **Berwa et al, 2022** | RMDT (at the PoC) followed by standard molecular test (if rapid test is negative) | Cobas Liat followed by RT-PCR Genexpert or RT-PCR R-DiaFlu or Respifinder (multiplex PCR) |
| | Standard molecular test | RT-PCR Genexpert or RT-PCR R-DiaFlu or Respifinder (multiplex PCR) |
| | Standard molecular test | RT-PCR Genexpert or RT-PCR R-DiaFlu or Respifinder (multiplex PCR) |
| **Bibby et al, 2022** | RMDT (at the PoC) | Cepheid Xpert Xpress Flu/RSV Assay (multiplex PCR) |
| | Standard molecular test | NxTAG® RPP (highplex PCR) |
| **Brooke-Pearce and Demertzi, 2019** | RMDT (at the PoC) | Cobas Liat influenza/RSV (multiplex PCR) |
| | Standard molecular test | Roche Flow Solution, FTD Resp 21 (highplex PCR) testing for Flu A and B/RSV |
| **Lankelma et al, 2019** | RMDT (at the PoC) | Cobas Liat (multiplex PCR) |
| | Standard molecular test | Lab PCR (multiplex PCR) |
| **Martinot et al, 2019** | RMDT (unclear site) | Alere i (isothermal NAAT) |
| | Standard molecular test | Lab PCR (multiplex PCR) |
| **Melhuish et al, 2020** | RMDT (at the PoC) | Cobas Liat (multiplex PCR) |
| | Standard molecular test | Lab PCR (highplex PCR) |
| **Peaper et al, 2021** | RMDT (at the laboratory) | Cepheid Xpert Xpress Flu/RSV Kit (multiplex PCR) |
| | Standard molecular test | RIDT, DFA alone or with lab PCR (duplex antigen) |
| **Wabe et al, 2019a** | RMDT (at the laboratory) | Cepheid Xpert Xpress Flu/RSV Kit (multiplex PCR) |
| | Standard molecular test | Allplex respiratory panels 1, 2, and 3 (highplex PCR) |
| **Wabe et al, 2019b** | RMDT (at the PoC) | Cepheid Xpert Xpress Flu/RSV Kit (multiplex PCR) |
| | Standard molecular test | Allplex respiratory panels 1, 2, and 3 (highplex PCR) |
| **Wesolowski et al, 2023** | RMDT (unclear site) | Cepheid Xpert Xpress Flu/RSV Kit (multiplex PCR) |
| | Standard molecular test | NxTAG RPP (highplex PCR) |
| **Yin et al, 2022** | RMDT (at the PoC or laboratory) | Cobas Liat (multiplex PCR) |
| | RADT | Antigen rapid tests and viral culture (singleplex antigen) |

[a]Multiplex PCR has between 3–5 molecular targets and highplex PCR has six or more molecular targets.

DFA, direct fluorescent assay; NAAT, nucleic acid amplification test; PCR, polymerase chain reaction; PoC, point-of-care; POCT, point-of-care testing; RAT, rapid antigen test; RADT, rapid antigen diagnostic test; RIDT, rapid influenza diagnostic test; RMDT, rapid molecular diagnostic test; RPP, respiratory pathogen panel; RSV, respiratory syncytial virus.

**Table 8. Baseline patient characteristics of studies testing for influenza virus.**

| Publication | Test | N | Age, median (IQR) | Male, n (%) | Comorbidities, n (%) |
|---|---|---|---|---|---|
| Au Yeung et al, 2021 | RMDT (at the PoC) | 122 | 69 (43–84) | 62 (51.0) | Charlson Comorbidity Index: 0 (0–2) |
| | Standard molecular test | 362 | 74 (51–84) | 178 (49.0) | Charlson Comorbidity Index: 1 (0–2) |
| Benirschke et al, 2019 | RMDT (at the PoC) | 63 (Flu A positive) | 41 (0.5–81.0)[a] | (39.7) | -- |
| | RMDT (at the PoC) | 51 (Flu B positive) | 32.8 (1.7–90.4)[a] | (47.1) | -- |
| | RMDT (at the PoC) | 128 (Flu A/B negative | 41.3 (2.8–94.5)[a] | (37.5) | -- |
| | RADT followed by standard molecular test (if RADT is negative) | 81 (Flu A positive) | 36 (5.1–88.3)[a] | (40.7) | -- |
| | RADT followed by standard molecular test (if RADT is negative) | 52 (Flu B positive) | 29.2 (2.6–93.7)[a] | (42.3) | -- |
| | RADT followed by standard molecular test (if RADT is negative) | 245 (Flu A/B negative) | 45.9 (0.7–86.3)[a] | (40.0) | -- |
| Berry et al, 2020 | RMDT (at the PoC) | 755 | -- | -- | Charlson Comorbidity Index: 4 (0–9) |
| | Standard molecular test | 390 | -- | -- | Charlson Comorbidity Index: 4 (0–7) |
| Berwa et al, 2022 | RMDT (at the PoC) followed by standard molecular test (if rapid test is negative) | 927 | 72 (49–85) | 476 (51.4) | -- |
| | Standard molecular test | 391 | 81 (65–87) | 176 (45.0) | -- |
| | Standard molecular test | 531 | 74 (55–85) | 268 (50.5) | -- |
| Bibby et al, 2022 | RMDT (at the PoC) | 220 | -- | 96 (48.0) | -- |
| | Standard molecular test | 222 | -- | 105 (47.3) | -- |
| Brooke-Pearce and Demertzi, 2019 | RMDT (at the PoC) | 376 | -- | -- | -- |
| | Standard molecular test | 121 | -- | -- | -- |
| Lankelma et al, 2019 | RMDT (at the PoC) | 624 (Patients with positive results) | 72 (58–82) | -- | -- |
| | Standard molecular test | 189 (Patients with positive results) | 76 (67–84) | -- | -- |
| Martinot et al, 2019 | RMDT (unclear site) | 72 (Patients with positive results) | 63 (25–92)[a] | 35 (48.6) | -- |
| | Standard molecular test | 106 (Patients with positive results) | 66 (22–95)[a] | 56 (52.8) | -- |
| Melhuish et al, 2020 | RMDT (at the PoC) | 204 | 65.5 (62.7–68.3)[b] | (39.7) | • Overall comorbidities: (90.2) • Chronic lung disease (49.0) |
| | Standard molecular test | 104 | 64.2 (60.0–68.4)[b] | (37.5) | • Overall comorbidities: (88.5) • Chronic lung disease (40.4) |
| Peaper et al, 2021 | RMDT (at the laboratory) | 3629 | 59.4 (20.5)[c] | 2069 (57.0) | -- |
| | Standard molecular test | 1489 | 60.2 (21.1)[c] | 890 (59.8) | -- |
| Wabe et al, 2019a | RMDT (at the laboratory) | 1209 | 77 (65–86) | 590 (48.8) | Charlson Comorbidity Index: 1 (0–2) |
| | Standard molecular test | 953 | 75 (62–84) | 473 (49.6) | Charlson Comorbidity Index: 1 (0–2) |

(*Continued*)

**Table 8.** (Continued)

| Publication | Test | N | Age, median (IQR) | Male, n (%) | Comorbidities, n (%) |
|---|---|---|---|---|---|
| **Wabe et al, 2019b** | RMDT (at the PoC) | 2250 | 69 (41–82) | 1103 (49.0) | -- |
| | Standard molecular test | 1491 | 53 (6–77) | 759 (50.9) | -- |
| **Wesolowski et al, 2023** | RMDT (unclear site) | 247 (Patients with positive results) | 17.6 (19.3)[c] | 120 (48.6) | Charlson Comorbidity Index: 0.25 (0.7) |
| | Standard molecular test | 33 (Patients with positive results) | 27.5 (25.8)[c] | 16 (48.5) | Charlson Comorbidity Index: 0.7 (1.6) |
| | RMDT (unclear site) | 88 (Adult patients with positive results) | -- | 33 (37.5) | • Diabetes mellitus: 5 (5.7) • Renal disease: 2 (2.3) • Coronary artery disease: 7 (8.0) • Chronic heart failure: 1 (1.1) • Asthma: 7 (8.0) |
| | Standard molecular test | 16 (Adult patients with positive results) | -- | 7 (43.7) | • Diabetes mellitus: 1 (6.3) • Renal disease: 0 (0.0) • Coronary artery disease: 2 (12.5) • Chronic heart failure: 1 (6.3) • Asthma: 1 (6.3) |
| **Yin et al, 2022** | RMDT (at the PoC or laboratory)/RADT | 293 | -- | 170 (58.0) | -- |

[a]Mean and range reported

[b]Median and CI reported

[c]Median and IQR reported.

IQR, interquartile range; PoC, point-of-care; RADT, rapid antigen diagnostic test; RMDT, rapid molecular diagnostic test.

molecular test, while two found similar rates compared to RADT followed by standard molecular test (if RADT is negative) and to standard molecular test. In Yin et al, 2022, intention to prescribe antibiotics did not change significantly after versus before the negative results for RMDTs (46.6% versus 50%), which was also the case for negative results for RADT (42.5% versus 48.1%).

Antiviral (oseltamivir) prescription among patients who tested negative was reported in six studies, all of which found a significant reduction in prescription rates with RMDT compared to the control cohorts. In Yin et al, 2022, negative results for RMDTs significantly reduced the intention to prescribe oseltamivir after versus before the test (1.7% versus 26.4%; p<0.0001); this was not reported for RADT. Of note, studies often described the patterns of prescriptions written after test results were made available, except Peaper et al, 2021 (analyzed the empiric prescription of oseltamivir) and Yin et al, 2022 (described clinician's intention to prescribe oseltamivir before and after the test).

**3.3.4. Influenza test positivity.** Influenza test positivity was similar between the RMDT and the control groups in half of the studies (note three studies only analyzed patients with positive test results, i.e., 100% positivity across all cohorts). There were imbalances between the RMDT and control cohorts in the other studies, including two studies with higher positivity rate in their RMDT cohorts (Wabe et al, 2019a [17.7% versus 1.3% (standard molecular test), respectively]), and Wabe et al, 2019b [35.1% versus 9% (standard molecular test)]).

**3.3.5. Length of stay in an emergency department.**   Median LoS at the ED in the overall population was reported in five studies, ranging from 3.7 to 11 hours with RMDT and from 3.8 to 11.9 hours with standard molecular test. Among patients who tested positive for influenza, median LoS at the ED ranged from 3.6 to 10.3 hours compared to 3.8 to 12.9 hours in patients undergoing RMDTs and standard molecular tests, respectively. Wesolowski et al, 2023 reported a shorter mean LoS at the ED with RMDT compared to standard molecular tests (3.4 versus 4.9 hours; p<0.01) in patients who tested positive for influenza.

**3.3.6. Admission to hospital.**   Rate of admission to hospital in all-comers was reported in three studies, all of which found significant reductions in hospitalization rates with RMDTs compared to standard molecular tests. Four studies reported admission rates within subgroups of patients with negative test results, of which three found significantly lower hospitalization rates in patients undergoing RMDTs compared to standard molecular tests; the other study also reported lower hospitalization rates (81.8% versus 95.3%) but did not conduct formal hypothesis testing. Specifically, Wabe et al, 2019b estimated the odds ratio (95% confidence interval) of hospitalization for standard molecular test versus RMDT (adjusting for baseline patient characteristics) to be 1.9 (1.6–2.3), 4.0 (2.3–6.8), and 1.5 (1.2–1.8) in all-comers, patients testing positive, and patients testing negative, respectively. In Yin et al, 2022, results of RMDTs did not have a significant impact on the physicians' intention to admit patients, which was also the case for RADT.

**3.3.7. Length of stay in hospital.**   LoS at the hospital in all-comers was reported in three studies. Melhuish et al, 2020 reported that mean LoS at the hospital was significantly shorter with RMDT compared to standard molecular tests (6.5 versus 11.5 days; p = 0.000), whereas Berry et al, 2020 did not find a significant difference between the two cohorts. Wabe et al, 2019a reported a median LoS of 4.2 days in both cohorts.

Three studies also reported LoS at the hospital among patients with negative test results. Brooke-Pearce and Demertzi 2019 reported a significant reduction in mean LoS at the hospital with RMDT compared to standard molecular tests (10.2 versus 16 days), whereas the difference in mean LoS was not significant in Wabe et al, 2019a and Berry et al, 2020.

**3.3.8. Ancillary testing.**   Berwa et al, 2022 reported significantly lower prescription rates of chest X-ray in patients undergoing RMDT (2018–2019 season) compared to those undergoing standard molecular tests in the 2016–2017 and 2017–2018 seasons (66% versus 82.3% and 79.4%, respectively; p<0.0001). This was also the case within subgroups of patients who tested positive (62.2% versus 86.0% and 80.7%; p<0.0001) and tested negative (67.6% versus 79.8% and 78.7%; p<0.0001) in that study. Conversely, Bibby et al, 2022 did not find a significant difference in this regard between patients receiving RMDTs and those receiving standard molecular tests. Lastly, Wabe et al, 2019b found that blood culture was prescribed significantly less often for patients undergoing RMDT compared to those undergoing standard molecular tests (56.0% versus 62.8%; p<0.001).

**3.3.9. Length of stay under medical observation.**   In Berry et al, 2020, the mean time on the open bay prior to isolation was significantly shorter in patients who tested positive with RMDT compared to those who tested positive with a standard molecular test (4.0 versus 20.9 hours; p<0.001) (Table 9).

## 4. Discussion

As of this study, in the US alone, influenza has resulted in 100,000–710,000 hospitalizations and 4,900–52,000 deaths each year since 2010 [73]. Global estimates from World Health Organization indicate that around seven million people have died from COVID-19 as of this report [3]. These fatal cases primarily resulted from severe forms of the disease accompanied by

**Table 9. Outcomes reported in studies testing for influenza virus.**

| Publication | Patient group | Test | N | Positive tests, n (%), p-value | LoS at the ED, hr, median (IQR) | Admission to the hospital, n (%) | LoS at the hospital, hr, median (IQR) | Ancillary testing, n (%) | Antimicrobial prescription in patients with negative test, n (%) | Time to test results, hr median (IQR) |
|---|---|---|---|---|---|---|---|---|---|---|
| Au Yeung et al, 2021 | Overall | RMDT (at the PoC) | 122 | 122 (100) | 6 (4.3–7.4)[a] | 89 (73.0) | 123.8 (65.3–196.3)[a] | -- | -- | 2.6 (2–3.8) |
| | | Standard molecular test | 362 | 362 (100) | 5.3 (3.6–10.6)[a], p = 0.90 | 252 (70.0) | 127.7 (69.1–264.2)[a], p = 0.72 | -- | -- | 22.9 (16.8–38.0), p<0.01 |
| Benirschke et al, 2019 | Overall | RMDT (at the PoC) | 242 | 114 (47.1) | -- | -- | -- | -- | -- | 0.48 (0.35–0.98)[b,c] |
| | | RADT followed by standard molecular test (if RADT is negative) | 378 | 133 (35.2) | -- | -- | -- | -- | -- | 0.27 (0.08–0.48)[b,c] 21.0 (5.0–28.0)[c] |
| | Flu A/B negative | RMDT (at the PoC) | 128 | 0 (0.0) | -- | -- | -- | -- | Antibiotic: 57 (44.5) Antiviral: 3 (2.3) | -- |
| | | RADT followed by standard molecular test (if RADT is negative) | 245 | 0 (0.0) | -- | -- | -- | -- | Antibiotic: 90 (36.7), NS Antiviral: 62 (25.3), p<0.005 | -- |
| Berry et al, 2020 | Overall | RMDT (at the PoC) | 755 | 164 (21.7) | -- | -- | -- | -- | -- | 2.9 (2.9–3.1)[c] |
| | | Standard molecular test | 390 | 90 (23.1) | -- | -- | -- | -- | -- | 31.2 (29.6–32.9)[c], p<0.001 |
| | Flu A/B or RSV positive | RMDT (at the PoC) | 164 | 164 (100) | -- | -- | 48.8 (52.6–64.7)[c] | -- | -- | 2.5 (2.2–3.0)[c] |
| | | Standard molecular test | 90 | 90 (100) | -- | -- | 64.3 (40.6–58.5)[c], p = 0.05 | -- | -- | 31.4 (28.0–35.2)[c], p<0.001 |
| | Flu A/B or RSV negative | RMDT (at the PoC) | 591 | 0 (0.0) | -- | -- | 75.2 (68.3–82.9)[c] | -- | -- | 3.0 (2.8–5.2)[c] |
| | | Standard molecular test | 252 | 0 (0.0) | -- | -- | -- | -- | -- | 29.8 (27.9–31.8)[c], p<0.001 |
| Berwa et al, 2022 | Overall | RMDT (at the PoC) followed by standard molecular test (if rapid test is negative) | 927 | 286 (29.9) | 11.0 (7.9–18.4)[b] | 516 (54.0) | -- | Chest X-ray: 631 (66.0) | -- | -- |
| | | Standard molecular test | 391 | 157 (39.3) | 11.9 (7.0–18.3)[b], p = 0.95 | 303 (75.8), p<0.0001 | -- | Chest X-ray: 329 (82.3), p<0.0001 | -- | -- |
| | | Standard molecular test | 531 | 186 (33.9) | 11.7 (7.5–18.6)[b], p = 0.95 | 338 (61.7), p<0.0001 | -- | Chest X-ray: 435 (79.4), p<0.0001 | -- | -- |
| | Flu positive | RMDT (at the PoC) followed by standard molecular test (if rapid test is negative) | 286 | 286 (100) | 9.6 (6.8–15.7)[b] | 128 (44.8) | -- | Chest X-ray: 178 (62.2) | -- | -- |
| | | Standard molecular test | 157 | 157 (100) | 11.5 (6.9–18.0)[b], p = 0.02 | 115 (73.3), p<0.0001 | -- | Chest X-ray: 135 (86.0), p<0.0001 | -- | -- |
| | | Standard molecular test | 186 | 186 (100) | 12.3 (7.7–19.8)[b], p = 0.02 | 102 (54.8), p<0.0001 | -- | Chest X-ray: 150 (80.7), p<0.0001 | -- | -- |
| | Flu negative | RMDT (at the PoC) followed by standard molecular test (if RMDT is negative) | 670 | 0 (0.0) | 11.7 (8.1–18.8)[b] | 388 (57.9) | -- | Chest X-ray: 453 (67.6), | Antibiotic: 243 (36.3) Oseltamivir: 1 (0.2) | -- |
| | | Standard molecular test | 243 | 0 (0.0) | 12.3 (7.1–19.1)[b], p = 0.42 | 188 (77.4), p<0.0001 | -- | Chest X-ray: 194 (79.8), p<0.0001 | Antibiotic: 117 (48.2), p<0.0001 Oseltamivir: 16 (6.6), p<0.0001 | -- |
| | | Standard molecular test | 362 | 0 (0.0) | 11.6 (7.5–17.3)[b], p = 0.42 | 236 (65.2), p<0.0001 | -- | Chest X-ray: 285 (78.7), p<0.0001 | Antibiotic: 172 (47.5), p<0.0001 Oseltamivir: 6 (1.7), p<0.0001 | -- |

(*Continued*)

**Table 9.** (Continued)

| Publication | Patient group | Test | N | Positive tests, n (%), p-value | LoS at the ED, hr, median (IQR) | Admission to the hospital, n (%) | LoS at the hospital, hr, median (IQR) | Ancillary testing, n (%) | Antimicrobial prescription in patients with negative test, n (%) | Time to test results, hr median (IQR) |
|---|---|---|---|---|---|---|---|---|---|---|
| **Bibby et al, 2022** | Overall | RMDT (at the PoC) | 200 | 47 (23.5) | 4.7 | -- | -- | -- | -- | 1.2 (0.76–2.8)[c] |
| | | Standard molecular test | 222 | 48 (21.6) | 3.9, p = 0.056 | -- | -- | -- | -- | 29.2 (19.4–54.7)[c], p<0.0001 |
| | Inpatients, Flu/RSV negative | RMDT (at the PoC) | 124 | 0 (0.0) | -- | -- | -- | Chest X-ray: (26.8) | Antibiotic: (56.5) Oseltamivir: (14.9) | -- |
| | | Standard molecular test | 140 | 0 (0.0) | -- | -- | -- | Chest X-ray: (27.1), p>0.99 | Antibiotic: (57.1) p>0.99 Oseltamivir: (27.5), p = 0.01 | -- |
| | ED, Flu/RSV negative | RMDT (at the PoC) | 30 | 0 (0.0) | -- | -- | -- | Chest X-ray: (9.1) | Antibiotic: (24.1) | -- |
| | | Standard molecular test | 34 | 0 (0.0) | -- | -- | -- | Chest X-ray: (11.1), p>0.99 | Antibiotic: (20.6), p = 0.77 | -- |
| **Brooke-Pearce and Demertzi, 2019** | Influenza A/B positive | RMDT (at the PoC) | 145 | 145 (100) | -- | 87 (60.0) | 232.8[a] | -- | -- | -- |
| | | Standard molecular test | 35 | 35 (100) | -- | 32 (91.4) | 314.4[a] | -- | -- | -- |
| | Influenza A/B negative | RMDT (at the PoC) | 231 | 0 (0.0) | -- | 189 (81.8) | 244.8[a] | -- | -- | -- |
| | | Standard molecular test | 86 | 0 (0.0) | -- | 82 (95.3) | 384[a] | -- | -- | -- |
| **Lankelma et al, 2019** | Overall | RMDT (at the PoC) | 1546 | 624 (4.0) | 3.7 (3.0–4.7) | (38.0) | -- | -- | -- | 1.0 (0.82–2.0)[b] |
| | | Standard molecular test | 591 | 189 (32.0) | 3.8 (3.0–4.7), p = 0.51 | (39.0), p = 0.23 | -- | -- | -- | 18.2 (13.0–22.0)[b], p<0.0001 |
| | Influenza positive | RMDT (at the PoC) | 624 | 624 (100) | 3.6 (2.9–4.6) | 455 (73.0) | 110.6 (65.3–191.0)[a] | -- | -- | -- |
| | | Standard molecular test | 189 | 189 (100) | 3.8 (3.2–4.6), p = 0.028 | 172 (91.0), p<0.0001 | 140.6 (96.0–270.5)[a], p<0.0001 | -- | -- | -- |
| | Influenza negative | RMDT (at the PoC) | 922 | 0 (0.0) | -- | (80.0) | -- | -- | Antibiotic: 364/743 (49.0) Oseltamivir: 4/667 (0.6) | -- |
| | | Standard molecular test | 402 | 0 (0.0) | -- | (93.0), p<0.0001 | -- | -- | Antibiotic: 229/375 (61.0), p = 0.0001 Oseltamivir: 20/400 (5.0), p<0.0001 | -- |
| **Martinot et al, 2019** | Overall | RMDT (unclear site) | 72 | 72 (100) | 10.3 (1.0–52.9)[b] | 28 (38.9) | 204 (48.0–792)[a,d] | Sputum culture positive: 1 (1.4) | -- | -- |
| | | Standard molecular test | 106 | 106 (100) | 12.9 (0.83–59.3)[b], p = 0.005 | 65, 61.3), p = 0.003 | 189.6 (24.0–696)[a,d], NS | Blood culture: 3 (2.8) | -- | -- |
| **Melhuish et al, 2020** | Overall | RMDT (at the PoC) | 204 | 85 (41.7) | -- | (74.5) | 155.8 (122.4–187.2)[a,d] | -- | -- | -- |
| | | Standard molecular test | 104 | 39 (37.5) | -- | (91.3), p = 0.000 | 275.8 (206.4–343.2)[a,d], p = 0.000 | -- | -- | -- |
| **Peaper et al, 2021** | Overall | RMDT (at the laboratory) | 3,629 | 812 (22.4) | 5.0[b] | -- | -- | -- | -- | 2.4[b] |
| | | Standard molecular test | 1,489 | 359 (24.1), p = 0.187 | 6.2[b], p<0.001 | -- | -- | -- | -- | 9.9[b], p<0.05 |
| | Influenza negative | RMDT (at the laboratory) | 2817 | 0 (0.0) | -- | -- | -- | -- | Antiviral: 48 (1.7) | -- |
| | | Standard molecular test | 1130 | 0 (0.0) | -- | -- | -- | -- | Antiviral: 44 (3.9) | -- |

*(Continued)*

**Table 9.** (Continued)

| Publication | Patient group | Test | N | Positive tests, n (%), p-value | LoS at the ED, hr, median (IQR) | Admission to the hospital, n (%) | LoS at the hospital, hr, median (IQR) | Ancillary testing, n (%) | Antimicrobial prescription in patients with negative test, n (%) | Time to test results, hr median (IQR) |
|---|---|---|---|---|---|---|---|---|---|---|
| Wabe et al, 2019a | Overall | RMDT (at the laboratory) | 1209 | 405 (17.7) | -- | -- | 101.8 (97.1–112.8)[c] | -- | -- | 2.3 (1.6–3.7) |
| | | Standard molecular test | 953 | 24 (1.3) | -- | -- | 100.9 (96.2–109.9)[c], p = 0.18 | -- | -- | 27.4 (23.0–36.8), p<0.01 |
| | Influenza A/B positive | RMDT (at the laboratory) | 440 | 405 (100) | -- | -- | 92.9 (81.5–98.6)[c] | -- | -- | -- |
| | | Standard molecular test | 210 | 24 (100) | -- | -- | 98.6 (94.5–116.9)[c], p = 0.01 | -- | -- | -- |
| | Influenza A/B negative | RMDT (at the laboratory) | 769 | 0 (0.0) | -- | -- | 116.2 (103.4–122.6)[c] | -- | -- | -- |
| | | Standard molecular test | 743 | 0 (0.0) | -- | -- | 101.4 (96.0–113.0)[c], p = 0.83 | -- | -- | -- |
| Wabe et al, 2019b | Overall | RMDT (at the laboratory) | 2250 | 790 (35.1) | 7.4 (5.0–12.9) | 1649 (73.3) | -- | Blood culture: 1259 (56.0) | -- | 2.4 (1.6–3.9) |
| | | Standard molecular test | 1491 | 134 (9.0) | 6.5 (4.2–11.9), p = 0.027 | 1159 (77.7), p<0.001[e] | -- | Blood culture: 937 (62.8), p<0.001 | -- | 26.7 (21.2–37.8), p<0.0001 |
| | Influenza/RSV positive | RMDT (at the laboratory) | 790 | 790 (100) | -- | 532 (67.3) | -- | -- | -- | -- |
| | | Standard molecular test | 134 | 134 (100) | -- | 105 (78.4), p<0.05[e] | -- | -- | -- | -- |
| | Influenza/RSV negative | RMDT (at the laboratory) | 1460 | 0 (0.0) | -- | 1117 (76.5) | -- | -- | -- | -- |
| | | Standard molecular test | 1357 | 0 (0.0) | -- | 1054 (77.7), p<0.05[e] | -- | -- | -- | -- |
| Wesolowski et al, 2023 | Patients with positive results | RMDT (unclear site) | 247 | 247 (100) | 3.4 (1.7)[f] | -- | -- | -- | -- | 3.5 (1.7)[f] |
| | | Standard molecular test | 33 | 33 (100) | 4.9 (2.1)[f], p<0.01 | -- | -- | -- | -- | 27.0 (6.6)[f], p<0.01 |
| Yin et al, 2022 | Overall | RMDT (at the PoC or laboratory) | 293 | 90 (30.7) | -- | -- | -- | -- | -- | 1.0 (0.88–1.3)[b] |
| | | RADT | 293 | 44 (15.0) | -- | -- | -- | -- | -- | -- |
| | PCR positive | RMDT (at the PoC or laboratory) | 90 | 90 (100) | -- | 41/78 (52.6) | -- | -- | -- | -- |
| | RADT positive | RADT | 44 | 44 (100) | -- | 18/37 (48.6) | -- | -- | -- | -- |
| | PCR negative | RMDT (at the PoC or laboratory) | 203 | 0 (0.0) | -- | 131/183 (71.6) | -- | -- | Antibiotic: 81/174 (46.6) Oseltamivir: 3/178 (1.7) | -- |
| | RADT negative | RADT | 249 | 0 (0.0) | -- | -- | -- | -- | Antibiotic: 90/212 (42.5) | -- |

[a]Calculated from days to hours, assuming one day = 24 hours

[b]Calculated from minutes to hours, assuming one hour = 60 minutes

[c]95% CI was reported

[d]Range reported

[e]Odds ratio (95% CI) of hospitalization for standard molecular test versus RMDT (adjusting for age and sex, as well as time, day, and mode of arrival at the ED) was 1.9 (1.6–2.3), 4.0 (2.3–6.8), and 1.5 (1.2–1.8) in all-comers, patients testing positive, and patients testing negative, respectively

[f]Mean and standard deviation were reported

CI, confidence interval; ED, emergency department; hr, hour; IQR, interquartile range; LoS, length of stay; NS, difference was not statistically significant; PCR, polymerase chain reaction; PoC, point-of-care; RADT, rapid antigen diagnostic test; RDT, rapid diagnostic test; RMDT, rapid molecular diagnostic test; RSV, respiratory syncytial virus.

cytokine storms [74, 75]. Until mid-November 2021, around 3.4 billion individuals were estimated globally to have been infected with SARS-CoV-2 at least once, with daily infections rising to 17 million in April, 2021 [76]. Timely diagnosis of viral respiratory infections with such heavy burden of illness has proven essential in preventing disease transmission, complications, hospitalizations, and deaths [77, 78].

Molecular diagnostic tests, such as PCR, have a prominent role in early detection of infectious diseases and are considered 'gold standard' for diagnosis of viral infections [13, 19]. With more efficient performance and the ability to be deployed at the PoC, RMDTs have become an important part of the process for rapid pathogen detection and on-site diagnosis [79, 80].

Results from this study showed that, compared to standard molecular tests, RMDTs reduce the rates of ancillary tests, such as chest X-rays and blood cultures, in patients presenting to the ED with COVID-19 and influenza-like symptoms. They also reduce unnecessary patient isolation, bay closures, exposure time for uninfected patients, and LoS in the ED and at hospitals, compared to standard molecular tests and RADTs. Studies also showed that RMDTs reduced use of antibiotics and oseltamivir in those with negative test results for influenza (with or without RSV). Lastly, RMDTs led to decreased time to test results and lower hospitalization rates in patients testing for SARS-CoV-2 and the influenza virus compared to standard molecular tests and RADTs.

RMDTs have been recommended over RADTs by the IDSA for detection of influenza viruses in respiratory specimens of outpatients [21], which aligns with the findings from this study, where RMDTs generally accelerated the diagnosis process and improved antiviral stewardship compared to other diagnosis modalities. It is noteworthy to mention that the included studies were all conducted in hospitals, EDs or other acute medical facilities. Patients in such healthcare settings benefit from RMDTs by avoiding long stays at medical facilities and, thereby, averting extended exposures to hospital pathogens, which can help decrease the likelihood of hospital-acquired infections and the resulting financial burden on healthcare systems. Results from the included studies should be interpreted with caution when extrapolating to the general population, which includes individuals with respiratory symptoms who may not have comorbidities or may not visit hospitals and EDs.

There was considerable heterogeneity in the included studies in terms of the design, objectives, baseline patient characteristics, and evaluated diagnostic tests. Studies were often observational and retrospective in design with variable quality and associated issues in terms of incomplete data collection and imbalanced distribution of prognostic factors across the analyzed cohorts. Such imbalances were, at times, highlighted and acknowledged. For example, the investigators in Brendish et al, 2020 mention that the longer LoS at the hospital in the RMDT group could be explained by the differences in the baseline respiratory signs and symptoms and NEWS2 scores between the RMDT and the standard molecular test cohorts. Due to the non-randomized nature of the study, patients in the RMDT group were recruited during the day by the research staff, and eligible patients were highlighted at the ED as being more likely to have COVID-19; therefore, patients with poor medical conditions may have been prioritized for RMDT by the clinical staff. However, potential imbalances remained unaddressed in most instances, e.g., when patient cohorts were systematically recruited across different influenza seasons and years, potentially leading to differences in baseline disease severity as well as level of supportive care and hospital services, subjecting the study results to selection bias.

Studies also used various diagnostic assays and devices. For example, while most studies used PCRs as their choice of RMDT, some used isothermal NAAT, which, despite some similarities (e.g., both amplify nucleic acid genetic material), is a different type of molecular diagnostic test that has been shown to be less sensitive than PCR for detection of SARS-CoV-2 [81]. Furthermore, some studies conducted RMDTs at the PoC (after training nurses), while

others used their existing laboratory staff to conduct the tests at the laboratories. Also, RSV was not exclusively tested for in any of the studies and was only investigated in addition to influenza in select studies (i.e., was not the focus of any of the included studies). Although the included studies consistently found improvements in most clinical aspects of the diagnosis process when using RMDT, regardless of the exact test site, time to test results are highly dependent on the overall workload of laboratory staff when RMDTs are used in laboratories and can be expected to reduce even further when these tests are conducted at the PoC [69]. Use of RMDTs at the PoC is also likely to increase the autonomy of the clinical staff at the ED in terms of workflow management and real-time adaptation of the indications, depending on factors such as patient bed availability [69].

This SLR provides an updated understanding of the most recent evidence on the clinical impact of RMDTs in patients with symptoms of viral respiratory infections, as compared to more conventional methods, such as standard molecular tests and RADTs. The results also confirm the findings from the previously published SLRs/MAs. In Brigadoi et al, 2022 [43], an SLR and MA were performed to investigate the impact of rapid and PoC tests for respiratory infections in children on antimicrobial prescriptions rates, length of stay, and other clinical and economic aspects of healthcare in high and low-middle income countries. The analyses showed an overall reduction in antibiotic prescriptions for RMDTs versus standard diagnostic tests (odds ratio = 0.59; 95% CI: 0.37–0.92) [43]. When comparing the rate of oseltamivir prescription for RMDTs versus standard tests, although the estimated odds ratio of 0.70 showed a reduction, the results were not statistically significant [43]. Weragama et al, 2022 [24] assessed, via an SLR, the clinical impact of rapid antigen and molecular diagnostic testing in children with respiratory symptoms presenting to ED. Although a formal MA was not conducted in the study, findings showed some evidence that antibiotic prescription and ancillary testing were reduced using RMDTs compared to other approaches, such as standard molecular test and immunofluorescence assay [24]. Lastly, Vos et al, 2019 [39] provided a qualitative summary of studies assessing the clinical impact of RMDTs for respiratory infections, which showed that turnaround times were significantly shorter with RMDTs compared with standard molecular tests. Other outcomes of interest that were explored (e.g., antibiotic and oseltamivir prescriptions, hospital admissions, LoS at the hospital) were reported in only a few of the included studies, which were often non-randomized and sometimes underpowered, leading to mixed evidence in terms of potential improvements with RMDTs over standard molecular tests [39].

The current SLR involved sensitive searches in peer-reviewed journals alongside additional searches of the proceedings of recent conferences, guided by pre-specified study eligibility criteria and following established guidelines. Despite these strengths, as the evidence base is continually developing, studies published after the search date or close to it (if not yet indexed) may not have been captured, which is a limitation applicable to all SLRs. Furthermore, in order to capture the most recent data pertaining to the research questions, searches were restricted to studies published in or after 2019; as such, there is a risk that relevant articles published earlier than 2019 were not identified. As discussed earlier, published SLRs and MAs had already investigated the diagnostic performance of RMDTs and other diagnostic tests. Therefore, the current SLR focused only on studies that compared these diagnostic tests in terms of clinical outcomes that, in the authors' opinion, were more patient-relevant, leading to a much narrower scope for the review. Since these outcomes were less frequently compared between the diagnostic tests of interest, only a few studies were ultimately included in our evidence base. Lastly, the search and selection were restricted to studies published in English. Therefore, there is a risk that non-English publications were not identified.

Overall, findings from this study suggest that RMDTs optimize patient flow and improve patient bed management by accelerating the patient characterization process and having a

positive impact on patient treatment and discharge decisions, thereby increasing isolation room availability and expediting access to hospital procedures, resulting in more timely clinical decision making. By reducing exposure time, these tests may also lower hospital-acquired infections and related costs. Lastly, RMDTs improve antimicrobial stewardship through lowering inappropriate use of antibiotics and antivirals in patients with negative test results and increasing appropriate oseltamivir use in patients at high risk of influenza complications.

## Supporting information

**S1 Table. Embase search strategies.**
(PDF)

**S2 Table. MEDLINE search strategies.**
(PDF)

**S3 Table. EconLit search strategies.**
(PDF)

**S4 Table. CENTRAL search strategies.**
(PDF)

**S1 Fig. Pooled results of risk of bias assessment for non-randomized studies (n = 20) using the ROBINS-I tool.**
(TIF)

**S2 Fig. Quality assessment of randomized controlled trials (n = 1) using the Cochrane Collaboration's tool for assessing risk of bias.**
(TIF)

**S1 Checklist. PRISMA 2020 checklist.**
(PDF)

## Author Contributions

**Conceptualization:** Ali Mojebi, Sam Keeping, Jordan G. Chase, Anne Beaubrun.

**Data curation:** Ali Mojebi, Ping Wu, Sam Keeping, Braden Hale, Jordan G. Chase, Anne Beaubrun.

**Funding acquisition:** Jordan G. Chase, Anne Beaubrun.

**Investigation:** Ali Mojebi, Ping Wu, Sam Keeping, Braden Hale, Jordan G. Chase, Anne Beaubrun.

**Methodology:** Ali Mojebi, Ping Wu, Sam Keeping, Braden Hale, Jordan G. Chase, Anne Beaubrun.

**Project administration:** Ali Mojebi, Sam Keeping, Jordan G. Chase, Anne Beaubrun.

**Supervision:** Ali Mojebi, Sam Keeping, Jordan G. Chase, Anne Beaubrun.

**Writing – original draft:** Ali Mojebi.

**Writing – review & editing:** Ali Mojebi, Ping Wu, Sam Keeping, Braden Hale, Jordan G. Chase, Anne Beaubrun.

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
