## [Decision Letter · Decision Letter 0]

3 Apr 2024

PONE-D-24-09537Clinical impact of rapid molecular diagnostic tests in patients presenting with viral respiratory symptoms: a systematic literature reviewPLOS ONE

Dear Dr. Chase,

Thank you for submitting your manuscript to PLOS ONE. After careful consideration, we feel that it has merit but does not fully meet PLOS ONE’s publication criteria as it currently stands. Therefore, we invite you to submit a revised version of the manuscript that addresses the points raised during the review process.

A rebuttal letter that responds to each point raised by the academic editor and reviewer(s). You should upload this letter as a separate file labeled 'Response to Reviewers'.• A marked-up copy of your manuscript that highlights changes made to the original version. You should upload this as a separate file labeled 'Revised Manuscript with Track Changes'.• An unmarked version of your revised paper without tracked changes. You should upload this as a separate file labeled 'Manuscript'.

We look forward to receiving your revised manuscript.

Kind regards,

Benjamin M. Liu, MBBS, PhD, D(ABMM), MB(ASCP)

Academic Editor

PLOS ONE

Journal Requirements:

   "I have read the journal's policy and the authors of this manuscript have the following competing interests: AM and SK are employees and shareholders of PRECISIONheor, which received funding from Cepheid for this work. PW and BH are employees of PRECISIONheor, which received funding from Cepheid for this work. JGC and AB are employees and shareholders of Cepheid, which provided funding for this work."  

We note that one or more of the authors are employed by a commercial company: PRECISIONheor

3. For studies involving third-party data, we encourage authors to share any data specific to their analyses that they can legally distribute. PLOS recognizes, however, that authors may be using third-party data they do not have the rights to share. When third-party data cannot be publicly shared, authors must provide all information necessary for interested researchers to apply to gain access to the data. (https://journals.plos.org/plosone/s/data-availability#loc-acceptable-data-access-restrictions) 

Additional Editor Comments:

Editor’s comments:

1. Abstract: “viral material” should be changed to “viral genetic material”. Statistical analysis should be added to the abstract. Conclusion should be re-written by removing the part lacking support from results, e.g., decreasing unnecessary use of antibiotics and antivirals, shift costs, or hospital-acquired infections.

2. “Decisions regarding choice of diagnostic tests are made based on the suspected pathogen, time, cost, availability of testing supplies, and patient risk category”. There is no reference to support this statement. More references need to be cited, with the following one as an example (citing suggestion is optional).

Laboratory diagnosis of CNS infections in children due to emerging and re-emerging neurotropic viruses. Pediatr Res. 2023 Dec 2. doi: 10.1038/s41390-023-02930-6. Epub ahead of print. PMID: 38042947.

3. Table 1: the comparators should not include antigen tests as the rapid molecular test has higher sensitivity than antigen tests. Please change other analysis and main text accordingly.

4.Why were only 22 studies included after the search? Did the authors miss any studies? Did the authors only use English? Please try to use other languages to re-run the selection process.

4. In discussion, “Global estimates from World Health Organization indicate that around seven million people have died from COVID-19 as of this report”. The authors should add severe COVID with cytokine storm are the main reason causing fatal cases. More references should be cited, including the following two references (citing suggestion is optional):

ref1: Role of Host Immune and Inflammatory Responses in COVID-19 Cases with Underlying Primary Immunodeficiency: A Review. J Interferon Cytokine Res. 2020 Dec;40(12):549-554. doi: 10.1089/jir.2020.0210. PMID: 33337932; PMCID: PMC7757688.

ref2: Clinical significance of measuring serum cytokine levels as inflammatory biomarkers in adult and pediatric COVID-19 cases: A review. Cytokine. 2021 Jun;142:155478. doi: 10.1016/j.cyto.2021.155478. Epub 2021 Feb 23. PMID: 33667962; PMCID: PMC7901304.

5. Given this study also covers RSV, the authors should introduce or discuss RSV.

Reviewers' comments:

Reviewer's Responses to Questions

**Comments to the Author**

1. Is the manuscript technically sound, and do the data support the conclusions?

Reviewer #1: Yes

2. Has the statistical analysis been performed appropriately and rigorously? 

Reviewer #1: Yes

3. Have the authors made all data underlying the findings in their manuscript fully available?

Reviewer #1: Yes

4. Is the manuscript presented in an intelligible fashion and written in standard English?

Reviewer #1: Yes

5. Review Comments to the Author

Reviewer #1: The manuscript accepted with minor revision

1- the manuscript table and figure must be written in journal style.

2- the Appendix figures and diagram must be in high resolution.

3- the references section ; author should select at least 50% of references from the last 5 years old.

4- The scores of the Rapid molecular diagnostic tests (RMDTs) should be described.

6. PLOS authors have the option to publish the peer review history of their article (what does this mean?). If published, this will include your full peer review and any attached files.

Reviewer #1: No

---

## [Author Response · Author response to Decision Letter 0]

22 Apr 2024

https://journals.plos.org/plosone/s/file?id=ba62/PLOSOne_formatting_sample_title_authors_affiliations.pdf 1) Filenames have been updated to: Fig1.tif, S1_Fig.tif, S2_Fig.tif, S1_Table.pdf, S2_Table.pdf, S3_Table.pdf, S4_Table.pdf, and S1_File.pdf.

2) Table formatting: removed black shades from the header rows and set the font size to 10 for table titles, content, legends and footnotes.

3) Unbolded Table and Figure cross-reference citations within the text.

4) Revised the filenames for the supplementary content and submitted separate files for appendix tables and figures

5) Updated how supplementary information was cited within the text (e.g., 'S1-S4 Tables')

We note that one or more of the authors are employed by a commercial company: PRECISIONheor

Please include both an updated Funding Statement and Competing Interests Statement in your cover letter. We will change the online submission form on your behalf. Our updated funding statement has been included in our Cover Letter to read: 

The funder (Cepheid) provided support in the form of salaries for authors JGC and AB, and consulting fees to PRECISIONheor. AM, PW, SK, and BH are employees of PRECISIONheor. The study concept and approach were informed by these commercial affiliations. All authors were involved in data collection, decision to publish, and preparation of the manuscript.

Please also provide an updated Competing Interests Statement declaring this commercial affiliation along with any other relevant declarations relating to employment, consultancy, patents, products in development, or marketed products, etc. 

Please include both an updated Funding Statement and Competing Interests Statement in your cover letter. We will change the online submission form on your behalf. Our Competing Interests Statement has been included in our Cover Letter to read:

"I have read the journal's policy and the authors of this manuscript have the following competing interests: AM and SK are employees and shareholders of PRECISIONheor, which received funding from Cepheid for this work. PW and BH are employees of PRECISIONheor, which received funding from Cepheid for this work. JGC and AB are employees and shareholders of Cepheid, which provided funding for this work. The authors do not have any other competing interests to declare. The competing interests declared above do not alter our adherence to PLOS ONE policies on sharing data and materials."

For studies involving third-party data, we encourage authors to share any data specific to their analyses that they can legally distribute. PLOS recognizes, however, that authors may be using third-party data they do not have the rights to share. When third-party data cannot be publicly shared, authors must provide all information necessary for interested researchers to apply to gain access to the data. (https://journals.plos.org/plosone/s/data-availability#loc-acceptable-data-access-restrictions) 

4) All necessary contact information others would need to apply to gain access to the data Thank you to the editor for providing additional clarification regarding this comment. Our Data Availability statement is updated to read: 

All data are available within the manuscript (Tables 1-9) or its Supplementary Information files. Hyperlinks are provided within the manuscript in the reference list

Abstract:

a) “viral material” should be changed to “viral genetic material”. 

b) Statistical analysis should be added to the abstract. 

c) Conclusion should be re-written by removing the part lacking support from results, e.g., decreasing unnecessary use of antibiotics and antivirals, shift costs, or hospital-acquired infections. Abstract:

a) “viral material” was updated to “viral genetic material”. 

b) The current study aimed to provide a qualitative summary of the recently published evidence by means of a systematic literature review. No statistical analysis was performed.

c) We have now revised the Conclusion section and emphasized reduced hospitalization rates and length of hospital stay as advantages of RMDT, which could potentially reduce patient exposure to nosocomial infections and their associated costs. In the Results section of the abstract, we had reported that RMDTs decreased frequency of unnecessary antiviral and antibacterial therapy, which supports the corresponding statement in the Conclusion section; therefore, no revision was made to that statement.

“Decisions regarding choice of diagnostic tests are made based on the suspected pathogen, time, cost, availability of testing supplies, and patient risk category”. There is no reference to support this statement. More references need to be cited, with the following one as an example (citing suggestion is optional).

Laboratory diagnosis of CNS infections in children due to emerging and re-emerging neurotropic viruses. Pediatr Res. 2023 Dec 2. doi: 10.1038/s41390-023-02930-6. Epub ahead of print. PMID: 38042947. Thanks for providing the suggested reference. We have added that reference along with two additional citations to support that statement.

Table 1: the comparators should not include antigen tests as the rapid molecular test has higher sensitivity than antigen tests. Please change other analysis and main text accordingly. We agree with the comment that RMDTs are more sensitive than antigen tests. In the Introduction section, we have emphasized this as well as the fact that RMDTs can also yield results in only 15-30 minutes. However, the objective of the current systematic review was to investigate how/if these test performance characteristics translate into actual clinical benefit for patients and healthcare providers in real-world clinical settings compared to antigen tests. This is the reason why antigen test was a comparator diagnostic method of interest for this review.

Why were only 22 studies included after the search? Did the authors miss any studies? Did the authors only use English? Please try to use other languages to re-run the selection process. In the Introduction section, we highlighted that published SLRs and MAs had already investigated the diagnostic performance of RMDTs and other tests. Therefore, the current SLR focused only on studies that compared these diagnostic tests in terms of clinical outcomes that, in our opinion, were more patient-relevant, leading to a much narrower scope for the review. Since these outcomes were less frequently compared between the diagnostic tests of interest in the published literature, only a few studies were ultimately included in our evidence base. We have now added a few statements in the Discussion section to emphasize the above. Lastly, we agree that we had to restrict our SLR to English publications as our reviewers were not able to review scientific literature published in other languages, which is a limitation that is arguably common to most SLRs. Please note we have mentioned this language restriction as a limitation of our SLR in the Discussion section.

In discussion, “Global estimates from World Health Organization indicate that around seven million people have died from COVID-19 as of this report”. The authors should add severe COVID with cytokine storm are the main reason causing fatal cases. More references should be cited, including the following two references (citing suggestion is optional):

ref1: Role of Host Immune and Inflammatory Responses in COVID-19 Cases with Underlying Primary Immunodeficiency: A Review. J Interferon Cytokine Res. 2020 Dec;40(12):549-554. doi: 10.1089/jir.2020.0210. PMID: 33337932; PMCID: PMC7757688.

ref2: Clinical significance of measuring serum cytokine levels as inflammatory biomarkers in adult and pediatric COVID-19 cases: A review. Cytokine. 2021 Jun;142:155478. doi: 10.1016/j.cyto.2021.155478. Epub 2021 Feb 23. PMID: 33667962; PMCID: PMC7901304. Thank you for the additional information. We have added a statement in this regard to the Discussion section, while citing the two suggested references. 

Given this study also covers RSV, the authors should introduce or discuss RSV. Thanks for flagging this. Unfortunately, RSV was not exclusively tested for in any of the included publications and was only investigated in addition to influenza in select studies (i.e., was not the focus of any of the included studies). We have now added a statement in the Discussion section to highlight this.

The manuscript table and figure must be written in journal style. 1) Table formatting: removed black shades from the header rows and set the font size to 10 for table titles, content, legends and footnotes.

2) Unbolded Table and Figure cross-reference citations within the text.

The Appendix figures and diagram must be in high resolution. Submitted separate high-resolution files for S1 Fig and S2 Fig

The references section; author should select at least 50% of references from the last 5 years old. This is already the case. Of the 81 citations referenced in our manuscript, 56 (69%) were published in or after 2020.

The scores of the Rapid molecular diagnostic tests (RMDTs) should be described. We have referenced the 'living systematic review and meta-analysis' study by Fragkou et al (2022) in our manuscript, which provides an up-to-date and detailed description of the diagnostic perforamance of RMDTs and antigen tests. As mentioned in the Introduction (and now also the Discussion) sections of our manuscript, our SLR intended to focus on the comparison of patient-relevant clinical outcomes between RMDT and other diagnostic tests, which had little overlap with studies like Fragkou et al (2022) in terms of objectives and scope. As such, we did not describe the performance aspect of these diagnostic tests in the current manuscript.

---

## [Editor Report · Decision Letter 1]

29 Apr 2024

Clinical impact of rapid molecular diagnostic tests in patients presenting with viral respiratory symptoms: a systematic literature review

PONE-D-24-09537R1

Dear Dr. Chase,

We’re pleased to inform you that your manuscript has been judged scientifically suitable for publication and will be formally accepted for publication once it meets all outstanding technical requirements.

Kind regards,

Benjamin M. Liu, MBBS, PhD, D(ABMM), MB(ASCP)

Academic Editor

PLOS ONE
---

## [Editor Report · Acceptance letter]

10 May 2024

PONE-D-24-09537R1 

PLOS ONE

Dear Dr. Chase, 

I'm pleased to inform you that your manuscript has been deemed suitable for publication in PLOS ONE. Congratulations! Your manuscript is now being handed over to our production team.

Kind regards, 

on behalf of

Dr. Benjamin M. Liu 

Academic Editor

PLOS ONE